# Neuromuscular and Neuromuscular Junction Manifestations of the PURA-NDD: A Systematic Review of the Reported Symptoms and Potential Treatment Options

**DOI:** 10.3390/ijms24032260

**Published:** 2023-01-23

**Authors:** Magdalena Mroczek, Stanley Iyadurai

**Affiliations:** 1Center for Cardiovascular Genetics & Gene Diagnostics, Foundation for People with Rare Diseases, 8952 Schlieren, Switzerland; 2Division of Neurology, Johns Hopkins All Children’s Hospital, 501 6th Ave S, St. Petersburg, FL 33701, USA

**Keywords:** PURA syndrome, motor symptoms, weakness, congenital myasthenic syndrome, 5q31.3 microdeletion syndrome, channelopathy

## Abstract

PURA-related neurodevelopmental disorders (PURA-NDDs) are a rare genetic disease caused by pathogenic autosomal dominant variants in the PURA gene or a deletion encompassing the PURA gene. PURA-NDD is clinically characterized by neurodevelopmental delay, learning disability, neonatal hypotonia, feeding difficulties, abnormal movements, and epilepsy. It is generally considered to be central nervous system disorders, with generalized weakness, associated hypotonia, cognitive and development deficits in early development, and seizures in late stages. Although it is classified predominantly as a central nervous syndrome disorder, some phenotypic features, such as myopathic facies, respiratory insufficiency of muscle origin, and myopathic features on muscle biopsy and electrodiagnostic evaluation, point to a peripheral (neuromuscular) source of weakness. Patients with PURA-NDD have been increasingly identified in exome-sequenced cohorts of patients with neuromuscular- and congenital myasthenic syndrome-like phenotypes. Recently, fluctuating weakness noted in a PURA-NDD patient, accompanied by repetitive nerve stimulation abnormalities, suggested the disease to be a channelopathy and, more specifically, a neuromuscular junction disorder. Treatment with pyridostigmine or salbutamol led to clinical improvement of neuromuscular function in two reported cases. The goal of this systematic retrospective review is to highlight the motor symptoms of PURA-NDD, to further describe the neuromuscular phenotype, and to emphasize the role of potential treatment opportunities of the neuromuscular phenotype in the setting of the potential role of PURA protein in the neuromuscular junction and the muscles.

## 1. Introduction

The *PURA* gene encodes Pur-alpha, a 322-amino-acid protein with repeated nucleic acid binding domains that are highly conserved from bacteria through humans. The *PURA* gene belongs to the PUR family, encoding four proteins and belonging to three different Pur (purine-rich element-binding proteins) protein classes: Pur-alpha (Pura), Pur-beta (Purb), and two forms of Pur-gamma (Purg) [1,2]. In humans, the conserved PUR domain is 55–70 amino acids in length, and three such PUR domains follow one after another. The first two PUR domains have nucleic acid binding properties, and the last one is responsible for PURA protein dimerization. The less-conserved protein binding regions of *PURA* (not containing PUR domains) have specific/unique protein binding properties, and show more variability among the species, as expected [3,4]. While *PURA* is ubiquitously expressed, *PURB* is expressed in the fibroblasts, myoblasts, and in myocardiocytes during heart failure [5], and *PURG* is expressed at low levels across the tissues [1]. 

In humans, variants and deletions encompassing *PURA* are associated with PURA-related neurodevelopmental disorders (PURA-NDDs), including the PURA syndrome and 5q31.3 microdeletion syndrome. The main difference between the two conditions is that 5q31.3 microdeletion syndrome is caused by a nonrecurrent genomic 5q31.3 deletion, which may encompass all or a part of *PURA* (10% of affected individuals; [6]), while PURA syndrome is caused by gene variants in the *PURA* locus. As expected, patients with 5q31.3 microdeletion syndrome are thought to have more severe symptoms due to the large nature of the deletions; the severity has been explained by the fact that the deletion may also include other nearby genes, such as *NRG2*, which is involved in the myelination of nerve sheaths [7,8]. Previously, all patients presenting with symptoms similar to that of PURA syndrome were classified as 5q31.3 microdeletion syndrome, as the disease had not been linked precisely to the *PURA* locus. However, now we understand that PURA syndrome, as opposed to the 5q31.3 microdeletion syndrome, is caused by selective PURA heterozygous pathogenic variants in a patient. PURA syndrome (MIM #616158) was first described in 2014 [9]. Gene variants solely in the *PURA* locus account for 90% of such PURA-NDDs [6]. Both diseases, PURA syndrome and 5q31.3 microdeletion syndrome, are very rare. Until now only 12 patients with 5q31.3 microdeletion syndrome have been reported (Appendix A), and [10] just over 475 individuals have been diagnosed with PURA syndrome, worldwide [11]. Homozygous pathogenic variants in the PURA gene have never been identified/described in humans.

### 1.1. Molecular Function of PURA and Its Role in the Central Nervous System and Other Tissues

PURA is a ubiquitous protein that has single-stranded RNA- and DNA-binding properties. It forms granules in the cytoplasm [12] and functions as a transcriptional activator of several proteins downstream. PURA homozygous knockout mice seem normal at birth, but soon develop neurological symptoms, such as what has been observed in patients with PURA syndrome around 2 weeks after birth [13]. Pura−/− mice have feeding problems and do not gain normal weight. Knockout Pura−/− mice were shown to have a significantly decreased neuron density in all three investigated regions of interest: cortex, cerebellum, and hippocampus, and had decreased synaptic density in hippocampal neurons, suggesting a role in neuronal transport, synapse formation, and memory formation [13]. There is no information regarding neuron and synapse density in other brain regions [13]. The knockout mice present with continuous and increasingly severe tremors upon motion, infrequent mobility, spontaneous seizures, and a flaccid tail. They are unable to walk normally, showing abnormal gait patterns with uncoordinated movements and weak hind limbs [13,14]. Heterozygous mice with PURA gene deletion (Pura+/−) present with no major phenotypic differences in comparison to wild-type mice, but show behavioral abnormalities, poor memory, abnormal gait, and limb and abdominal hypotonia [13,14,15]. In the pathomorphological examination, heterozygous Pur-alpha mice present with a significantly reduced number of neurons and dendrites in two out of four investigated brain regions: the cerebellum and hippocampus. No significant differences have been reported for the amygdala and prefrontal cortex [15]. PURA plays a critical role in neuron formation and is expressed most intensively during 5 to 15 days of a mice’s life in the time of the most obvious neuron formation [13]. Developmental NMJ synapse elimination in mice occurs during the first two weeks postnatally. In the sternomastoid muscle at birth, 90% of cells were multiple innervated at birth, a little less than half of them already single innervated at P7, and all of them single innervated at P14 [16]. Based on this information, the possibility of PURA’s involvement in the pruning of NMJ synapses or synapse elimination as a functional role for PURA cannot be excluded and, in fact, is deemed more likely. 

The role of PURA in the central nervous system (CNS) has been well elucidated from a molecular standpoint [2]. In the CNS, PURA regulates two major structural CNS proteins. One is the PLP1 protein [17], which is essential for myelination and binds to the promotor region of the myelin basic protein (MBP) in oligodendrocytes [18]. In the mass spectroscopy study confirmed by Western blot to assess the oligodendral localization, both Pur-alpha and Pur-beta ranked as top proteins interacting with the PLP1 protein [17]. This molecular interaction is in line with the demyelination observed in patients. PURA, together with mRNAs and transcription factors, forms RNA granules that are transported inside neurons thanks to the binding to KIF5 [19]. Pur-alpha-positive granules are localized in both the shafts and spines of dendrites from the cultured hippocampal cells [18,20]. Pur-alpha binds with the key protein for the neuronal transport, such as zipcode-binding protein 1 (ZBP1), fragile X mental retardation protein (FMRP), cytoplasmic polyadenylation element-binding protein (CPEB), survival of motor neuron (SMN), hnRNPA2, and purine-rich element-binding protein-a (Pura) [21].

Pur-alpha interacts not only with cytosolic/nuclear proteins, but also with nuclear RNAs. Pur-alpha in the so-called paraspeckles, which are nuclear membraneless compartments, forms an interactome with NEAT1 and MALAT1 nuclear RNAs in this localization [22]. A receptor interacting with PURA in the brain is metabotropic glutamate receptor 5 (mGluR5). mGluR5 in the hippocampus is involved in the synaptic plasticity through oscillations in the long-term potentiation in mice studies [23]. mGluR5 is impaired in several neurodevelopmental and psychiatric disorders [24]. Although PURA gene deletion may play a role in several diseases [25,26], one focus of investigations related to PURA lies in the realm of neurodegenerative expansion disorders, given its role in the regulation of transcription and translation [27]. Together with other proteins, PURA forms density granules in diseases related to the RNA repeat expansion disorders, such as fragile X mental retardation protein (FMRP), encoded by *FMR1* gene [19,28] and amyotrophic lateral sclerosis (ALS)/frontotemporal dementia (FTD) spectrum disorder [26,29]. Several potential mechanisms of PURA involvement in these diseases have been hypothesized. One theory is that the expanded rGGGGCC hexanucleotide repeats could sequester specific RNA-binding proteins, such as PURA, taking PURA away from its normal functions—which ultimately leads to cellular death and, consequently, neurodegeneration [28], as has been suggested in ALS and FTD [29]. As a test of this hypothesis, in zebrafish and cell culture models of ALS, Pur-alpha bound directly to the toxic RNA repeats and overexpression of PURA prevented the axonopathy in a dose-dependent manner [30]. 

### 1.2. PURA’s Role in the Muscle

PURA is present in nearly all tissues, including muscle, and appears to be involved in controlling orchestrated gene expression in myocytes. PUR proteins often act as negative regulators of gene expression. PURA and PURB suppress alpha- and beta-myosin heavy chains [5,31] by regulating translational and transcriptional factors binding properties to RNA and DNA. This is especially pronounced during myocyte differentiation and leads to the interplay between repression and activation of the muscle myosin-heavy chain (Mhc) transcription. An interaction of PURA and PURB with circSamd4, one of the circrRNA, is necessary for the binding with MHC promoter during the differentiation of myotubes [32]. PURA and PURB also regulate a heart-specific expression of MHC alpha by binding to the DNA elements. MHC alpha is downregulated in a PURA-dependent manner in heart failure as well [5]. In the myofibroblast during wound healing, a dynamic interplay between PURA, PURB, and transcriptional activators and repressors has been reported. PURA stabilized an interaction between the promoter and repression complex, whereas PURB was more effective in disrupting it [32]. In the mature myofibroblast, PURA replaced PURB. The aforementioned target mGlu5R has been suggested to play a role in the peripheral nervous system, among others, in pain transmission [33]. The Pur-alpha has been suggested to accumulate in spines secondary to the postsynaptic mGlu5-dependent cascade [20]. However, in the peripheral nervous system PURA localization has never been confirmed with staining, although electrophysiological studies suggest mixed pre- and postsynaptic localization [34]. The protein localization and its interaction partners in the synapses could help identify potential treatment options. 

In humans, the clinical aspects of central nervous system defects in patients with PURA syndrome and 5q31.3 microdeletion syndrome have been well described [35]. The main symptoms are neurodevelopmental delay, learning disability, and epilepsy. Consequently, brain MRIs from most PURA syndrome patients show abnormalities, including variable myelination delay. However, pathological evaluation of the brain from one patient showed no major abnormalities, except for signs of chronic inflammation [36]. The role of PURA in the neuromuscular system is only beginning to be understood.

In this manuscript, we review the published information on the motor symptoms in PURA- related neurodevelopmental disorders, the origin of the weakness, and the potential treatment options based on what is known about PURA’s function in the muscle/neuromuscular junction. 

## 2. Materials and Methods

To review the motor symptoms and weakness in the PURA syndrome, the PubMed database was searched on 21.11.2022, with the keywords “pura AND syndrome” (93 hits) and “pura AND 5q31.3 microdeletion syndrome” (0 hits). The searching and reviewing strategy were in line with the PRISMA criteria [37]. The initial search produced 93 hits (Figure 1). Only manuscripts with full text available were considered (59 hits). All 60 manuscripts were manually reviewed by two independent researchers (M.M. and S.I.). Thirty-three manuscripts were not relevant to the topic, leaving twenty-seven describing the symptoms of PURA syndrome. Two manuscripts [35,38] partially reported patients that already had been described. P2, reported in [38], had already been described in [39]. However, we included P1 from this manuscript and the manuscript in the discussion regarding therapy. Johanessen et al. [35] included 142 patients; only these 67 reported for the first time have been included here. Th article inclusion workflow is illustrated on Figure 1. For the scope of this review, we included both patients with PURA syndrome and 5.11 microdeletion syndrome. A total of 27 studies were included in this systematic review, with a largest one including 67 individuals and 17 single case reports. All included manuscripts are listed in the Appendix A. 

The following features that could be related to both central and peripheral weakness were extracted and analyzed separately (Appendix A) in all included patients: muscular hypotonia, neurodevelopmental delay, respiratory insufficiency with CPAP necessity, issues related to feeding, myopathic/atonic face, ptosis, long face, deep tendon reflexes (DTR), muscle biopsy, and electrophysiological tests (nerve conduction velocity (NCV) and electromyography (EMG)). They were extracted from the original manuscripts and entered manually in table format: Yes—present; No—not present; NA—not reported/not performed/no info; and, in case of EMG, muscle biopsy and NCV/EMG normal or abnormal. For quantifying the relative prevalence/frequency of neuromuscular symptoms in patients with PURA syndrome, we counted the symptom occurrence as a unit count, based on the description given. If there was no information about the symptom or it was described as negative, the unit count was zero.

## 3. Results and Discussion

We included altogether 193 PURA-related neurodevelopmental disorder (PURA-NDD) patients; among them, 10 with 5q31.3 microdeletion syndrome and 183 with point variants in *PURA* locus only from 27 studies [6,7,8,9,34,35,36,38,39,40,41,42,43,44,45,46,47,48,49,50,51,52,53,54,55,56,57]. The biggest study contained 67 participants, and there were 17 single case reports included. The workflow is presented on Figure 1. All studies included are listed in the Appendix A. 

### 3.1. Motor and Neuromuscular Symptoms in PURA Syndrome

Hypotonia is a general term used to describe a diminished tone and can result from the abnormality of the central nervous system (central hypotonia), peripheral neuromuscular system (peripheral hypotonia), or both (mixed hypotonia). Usually, it is thought that some symptoms can suggest the type of hypotonia. Impaired cognitive development, seizures and hyper-reflexia feature central hypotonia, whereas decreased DTRs, lack of antigravity movement, and muscle atrophy together with a normal cognitive function, point toward a peripheral hypotonia associated with neuromuscular weakness [58].

Neonatal weakness, described also as muscle hypotonia or “floppy infant”, is the most consistent motor symptom in PURA syndrome and appears in 98% of patients reported (Table 1). In four patients, no hypotonia was reported ([9], p. 2; [39], p. 32; [8], p. 11; and p. 15). Patients with PURA syndrome usually displayed mild-to-moderate weakness, although in some cases, severe weakness was described. During development, weakness associated with PURA syndrome usually mildly improves, especially after 1 year of life, however it may persist. Until now, the weakness in PURA syndrome used to be categorically assumed to be weakness of central origin. Such an assumption was in line with reported brain myelination defects and PURA’s purported role in the CNS [36,39]. According to the Lee et al. [36], a diagnosis of cerebral palsy was frequently assigned to PURA patients because of the increased tone in lower extremities. In accordance with CNS involvement, in seven cases, there was an additional spastic weakness component, existing parallel to the hypotonia. The oldest patient (26 years old) with a 5q31.2q31.3 deletion showed a significant tetraparesis in the end disease stage [41]. Additional features, such as positive Babinski reflex in two cases [39], ataxia [6,9,36], dystonia [9,55] and choreoathetosis [55] confirmed CNS involvement. DTRs have been described as brisk in three cases, again suggesting a CNS involvement.

However, the neuromuscular and synaptic weakness have been more appreciated recently (Figure 2). Especially, in a few cases, the weakness has been described as very severe (paralytic) (e.g., [51], p. 2, [34]). These children were tested at the first instance for SMA in the neonatal period but resulted negative. There were no elicitable DTRs, and respiratory insufficiency requiring artificial ventilation at birth or shortly after birth has been reported [51]. In the case of one patient ([39], p. 25; [38], p. 2—the same patient described twice), the weakness was reported as proximally pronounced instead of usually described generalized weakness. This patient had a repetitive electrostimulation suggestive for CMS and has been described in detail in [38] with prevalent symptoms of a synaptic disorder. In another patient ([38], p. 1), weakness was reported as fluctuating, again suggesting a neuromuscular junction disorder.

In most of the cases, diminished DTR has been reported [34,51]. In three cases there were either no reflexes or weak reflexes that during the disease course progressed to normal [7] or hyperreflexia [41] respectively. In one case, fluctuating DTRs have been noted [34]. In three cases, the DTR were normal. In most cases, there was no information regarding reflexes reported (173 out of 193 patients included).

Another feature in favor of peripheral components of weakness is the myopathic facies, described also as an atonic face. The mouth is open, there is a high-arched palate and, in a few patients, mild ptosis, sometimes fluctuating, is noted [39]. Although the face is most commonly described as atonic/myopathic in PURA patients, some of the affected children have additional dysmorphological features, including a high anterior hairline, almond-shaped palpebral fissures, and full cheeks; however, none of the features, not even a myopathic face, were consistent among all PURA syndrome patients. It is very difficult to identify the cases without myopathic face, as no information about the presence of myopathic face may result either from lack of the symptom or from lack of examination of this particular feature. In the manuscript of Reijnders [39], dysmorphologic aspects of faces have been evaluated in detail. Fifty-three patients presented with an atonic/myopathic face and in thirty-one patients it was explicitly stated that there was no myopathic face. Additionally, seven patients presented with unilateral [7] or mild bilateral ptosis [39]. One patient developed cardiomyopathy in the neonatal age [45]. In three cases where creatine kinase was available, it was mildly elevated in two, (191; ULN: 141 U/L ([48], p. 1), and 1600; ULN140 U/L ([48], p. 2)) and normal in one ([51], p. 3).

Only few individuals reached ambulation. The ambulation has been reached between the age of 1 and 12 years [5,35,42,43]. The lack of ambulation was attributed to a variety of factors, weakness often being most significant. Out of 193 patients described, only 73 patients could walk. The gait was described to be broad basic [9] or ataxic [6,9]; [36], p. 5; [39], p. 26, [48], p. 1 or toe walking ([39], p. 15) or stooped, diplegic ([39], p. 12). Some patients needed a walker (e.g., ([39], p. 23)) or had frequent falls ([36], p. 7 and p. 13). The ataxic gait improved spontaneously in one case ([51], p. 1). In some patients, worsened walking ability was noted after the onset of seizures (e.g., [36], p. 14). In addition, in others, a walking regression of unknown origin was noted [35].

A total of 124 patients had respiratory distress. This includes both central and peripheral (obstructive) apnea in the neonatal period. In the majority of cases, apnea was reported as “apnea” in general and was not classified as central or peripheral. Often there was both a central and peripheral component of the apnea [36,47]. In total, at least four patients needed CPAP ventilation, usually only temporarily. Fifty-three patients required a ventilatory support: in twenty-four cases mechanical ventilation and in ten cases oxygen. Apnea is often resolved after the first year of live with only exceptional cases requiring CPAP after the first year of life. However, during a systemic infection, a first manifestation or a short-term recurrence of apnea was reported (e.g., [36], p. 7; [47]).

### 3.2. Electrophysiological and Morphological Studies

Nerve conduction velocity (NCV) and/or EMG data were available for 15 patients. Out of these three cases revealed myopathic findings, and one case was reported to be normal. In three cases, there were myasthenic features were noted in electrodiagnostic studies. In one patient it was abnormal decrement was noted in repetitive nerve stimulation studies (RNSs) [39]. The same case was described in greater detail in Qashqari et al. (2022) [38]. The patients showed double peak compound muscle action potential (CMAP) plus abnormal RNS with a decremental response (49%) between the first and fourth stimulus with 3 Hz low-frequency RNS without significant increment response to 20 Hz high-frequency RNS. This suggested a slow channel pathology. Another patient showed a decremental response (46%) with 3 Hz low-frequency RNS and initial decrement (76%) followed by rebound increment (50%) at 20 Hz high-frequency RNS. The third case with a decrement/increment response showed features of the mixed pre- and post-synaptic involvement described [34]. In one case, fast (30 Hz) repetitive nerve stimulation showed a small, nonsignificant increment suggestive of potential pre-synaptic involvement. EMG results were in line with an irritable myopathy and revealed increased insertional activity, several low-amplitude, normal duration motor unit action potentials (MUAPs), and simple repetitive discharges (SRDs) (at ∼40 Hz) in a distal leg (tibialis anterior) muscle. EMG of same muscle showed increased insertional activity, 1+ Fibs/positive sharp waves (PSWs), few low-amplitude MUAPs, and normal recruitment patterns. EMG of the left vastus medialis was essentially normal, except for a few low-amplitude, small-duration MUAPs. In one case, electrical myotonia, positive sharp waves, and fibrillations with increased insertional activity in the first dorsal interosseous and the tibialis anterior [47] were noted. Most of the patients have not been investigated with RNSs, so synaptic involvement could not be commented on in those cases. Nerve conduction velocities were normal in most cases; however, a motor- or sensorimotor neuropathy was reported in five individuals at young age [39] In seven cases, the EMG/NCV were reported to be normal. 

Muscle biopsy was performed altogether in 15 cases. In four cases, muscle fiber type disproportion was reported, varying from the diagnosis of congenital fiber type disproportion to the mild, nonspecific excess in fiber type. In one case, diffuse muscle fiber atrophy was described, but not further information was available [42]. In two cases there was prevalently an atrophy of type 2 muscle fibers [51]. Myosin staining was reported only in two cases: in one case, an increased neonatal myosin [34]; and in the second case, mild atrophy of the fast, myosin-containing muscle fibers were observed [51]. In the H and E staining, muscle fibers showed occasional central nuclei. In one case immature muscle fibers were reported—usually noted as small muscle fibers and with increased neonatal myosin-positive muscle fibers. In the case with a clinical and electrophysiological presentation similar to that of a myasthenic syndrome, light microscopy level did not reveal any tubular aggregates [34]. In three cases, muscle biopsy was reported as normal [39].

### 3.3. Limitations

Given the predominant CNS symptoms associated with PURA-NDD, neuromuscular/synaptic symptoms were not adequately examined and reported in earlier studies. However, we have unearthed several aspects from previously reported literature that could be attributed to neuromuscular/NMJ deficits. Even so, due to the nature of this retrospective/metadata analysis, our study has several limitations and we would like to acknowledge those. In terms of the classification of weakness as either central or peripheral, it should be remembered that not all patients underwent a detailed clinical examination or it has not been reported, although most likely performed. Full examinations were not reported in many prior reports. DTR has been described only in 15 cases usually evaluated only once and maximum two times during the clinical examination (per provided report). Therefore, DTR fluctuation has not been assessed systematically. In addition, in several reported cases, according to current, local standard of practices, invasive studies such as EMG and muscle biopsies have not been performed. Moreover, when the neuromuscular physical examination has not been performed or has been performed, but not described, the symptom has often been counted as not present in the current study. The second limitation is the fact that different clinicians examined the patients at the different developmental stages. Especially in case of the PURA syndrome, where the neurodevelopment is very delayed, it is not possible to predict the milestones that will not be reached. The follow up has usually not been reported, although it is to be assumed that most of the patients underwent follow up. Last, but not least, with our search strategy, we were not able to encompass all PURA-related neurodevelopmental syndrome patients. For example, up to date 12 patients with 5q31.3 microdeletion syndrome have been reported so far. Two of them were not included to our manuscript as they were not identified with our search strategy in PubMed with the keywords used [10].

### 3.4. Treatment 

There is no FDA-approved treatment modifying the course of PURA syndrome, either symptomatic or disease-modifying. However, at least for the muscular/neuromuscular junction presentation of PURA syndrome, an empirical treatment has been tried in three cases. One drug used for the treatment in all three treated cases was pyridostigmine. Pyridostigmine inhibits the action of acetylcholinesterase and is generally used for symptomatic treatment of symptoms in patients with myasthenia gravis. In all three cases associated with NMJ disorder, the patient was started on pyridostigmine. One patient showed clinical improvement [34]. The patient was started on pyridostigmine every 6 h on the 64th day of life with a notable improvement in tone and apneic events within 24 h of initiation with apparent weaning off a response before the administration of the next dose, without any untoward consequences. No major side effects, such as bradycardia or increased secretions were reported. The child continues to show improvement in muscle weakness (unpublished results). In two other cases, pyridostigmine therapy has been started as well. In one patient treatment with pyridostigmine was ineffective and was associated with worsening respiratory status. The patient experienced several episodes of profound apnea and bradycardia (heart rate into the forties), which required stimulation for resolution of the bradycardia. This patient was consequently started on salbutamol, which improved the symptoms of neurotransmission deficit [38]. The treatment correlated with resolution of apneic spells, a reduced need for respiratory support (no longer required NIPPV), and subjective improvement in terms of his extremity strength. In the third patient, therapy was stopped after several days as it failed to show clinical benefit [38]. 

To date, no genotype–phenotype correlations have been established in formal studies. It has been hypothesized that pathogenic variants located in the third PUR domain cause more severe symptoms, however definitive evidence is still lacking [39]. Dai et al. [8] found not significant genetic-phenotypic correlation between 5q31.3 microdeletion syndrome and PURA syndrome, apart from more dysmorphic facial features in 5q31.3 microdeletion syndrome. To the best of our knowledge, no significant genotype-phenotype correlations have been established, including potentially hypomorphic variants [35,36,39].

Treatment response to treatment in PURA patients may be due to the pathological variants affecting different areas/locations within the protein structure (affecting differential domains of the tertiary structure). The patient reported in Wyrebek et al. 2022 [34] carried the c.697_699 (p.Phe233del) variant that is a recurring variant located in the PUR-III domain. The patients reported in Qashqari et al. 2022 [38] had variants affecting the PUR-I (c.313_318del; p.Ser105_Val106del) and the PUR-II domains (c.616_618delATC; p.Ile206del), and these patients did not respond to pyridostigmine. As of now, little is known about the localization of the PURA protein in the NMJ and how different variant genotypes are related to the phenotype. While it is tempting to conclude that treatment may be dependent on the affected protein domain structure, it is hard to assign treatment response based on functional domain involvement at this time and awaits further studies. Neuromuscular symptoms have not been sufficiently systematically described so that no genotype–phenotype between them and genetic variants is possible.

Salbutamol and pyridostigmine work with different mechanisms of actions—and salbutamol has been used successfully in a broader range of patients with neuromuscular diseases, rather than specific NMJ disorders, as with pyridostigmine as an acetylcholinesterase inhibitor. To quote an example, central core disease, which is not deemed to be a true NMJ disorder (rather a channelopathy), is treated successfully with salbutamol, although the exact mechanism of action of this drug remains unclear. Other therapies with pyridostigmine are ongoing, and the preliminary results are promising (unpublished observations; SI, MM). It is also not clear whether other NMJ channel modulators, such as 3,4-diaminopyridine (3,4-DAP) or 4-aminopyridine (4-AP), might have a beneficial role in treatment of PURA patients.

Another treatment that may be considered is fluoxetine. Fluoxetine is a selective serotonin re-uptake inhibitor. It is administered for a depression, obsessive compulsive disorder or eating disorder treatment. The properties of fluoxetine have also been investigated in the neuromuscular junction. Fluoxetine blocks the neuromuscular transmission and increases the potency of rocuronium-induced neuromuscular block [59]. Slow channels myasthenic syndromes are related to the prolongation of channel opening in the acetylcholine receptors [60]. The treatment of the slow channel CMS with fluoxetine is a standard [61]. It was suggested that fluoxetine interacts with a nicotinic receptor in the not depolarized mechanism. Fluoxetine can cause mild to severe adverse effects: among them nausea, nervousness, insomnia, anorgasmia, and hyponatremia in the elderly and increased suicide risk in depressed individuals [61]. Since some of the patients with PURA syndrome show feature of slow channel CMS, we hypothesize that in such cases fluoxetine may be trialed as well.

Last but not the least, there is no attempted treatments for the central component of the PURA-NDD. Epilepsy can be usually sufficiently controlled with drugs. Given that that the synaptic involvement and in CNS might be considered a channelopathy, 4-aminopyridine, could be considered a therapeutic option. The drug has already been trialed in the patients with gain-of-function KCNA2-encephalopathy and has been approved in multiple sclerosis for the treatment of ataxia. Two out of six patients experiencing generalized tonic-clonic seizures showed marked improvement, three showed no effect, and one worsening [62] in treatment with 4-aminopyridine. However, the possibility of epileptic seizures should be kept in mind, especially in PURA related neurodevelopmental disorder patients, who often eventually develop seizures.

## 4. Conclusions

Peripheral or mixed pre- and postsynaptic weakness can contribute significantly to the clinical picture of the motor aspects of PURA syndrome amid fairly well-known aspects of the central nervous system deficits. The contribution of the peripheral and neuromuscular junction component is variable and can fluctuate spontaneously. The peripheral or mixed pre- and postsynaptic component is not a consistent feature and is present only in some individuals to the varying degrees, as can be seen in neuromuscular junction disorders. PURA syndrome, at least in some cases, can manifest as a channelopathy, in a broader sense, with a specific role in NMJ, akin to CMS. The hypothesis is supported by EMG/NCV findings and treatment response to NMJ modulators. The treatment with either pyridostigmine or salbutamol proved to be effective in two cases, confirming a neuromuscular junction component of the weakness. Treatment with fluoxetine and other NMJ modulators may be considered in older children, as it is approved for children aged eight and older.

While localization of PURA in the CNS has been done in model organisms in many exquisite studies, no single study has tried to localize PURA in human muscle/neuromuscular junction, although implicated. The role and localization of PURA in the synapses may influence not only the treatment of peripheral symptoms but may also be the key to treatment of cognitive aspects of PURA-related neurodevelopmental disorders.

## Figures and Tables

**Figure 1 ijms-24-02260-f001:**
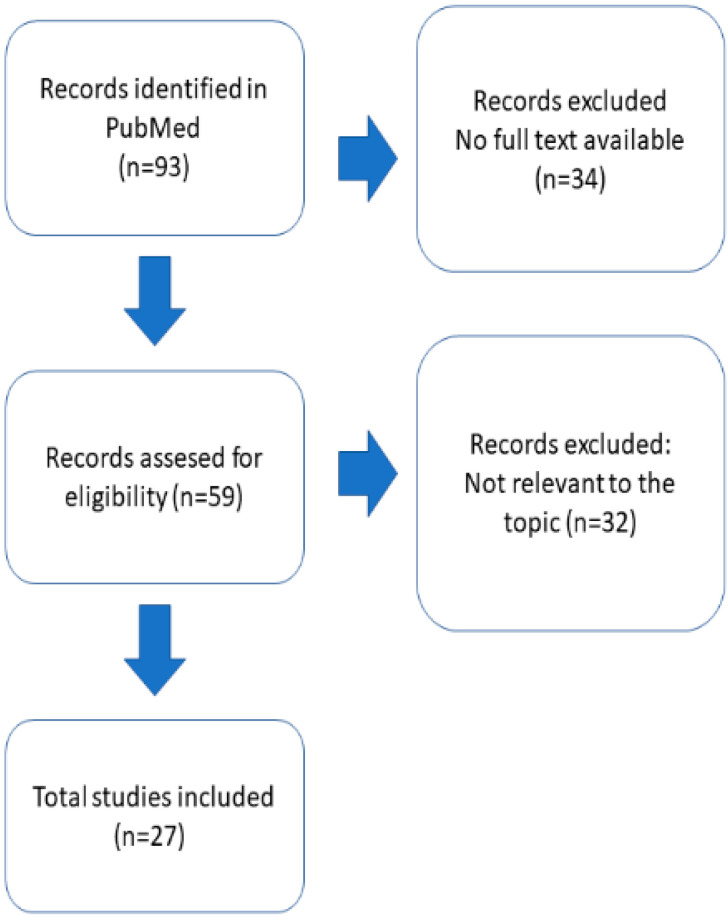
PRISMA workflow.

**Figure 2 ijms-24-02260-f002:**
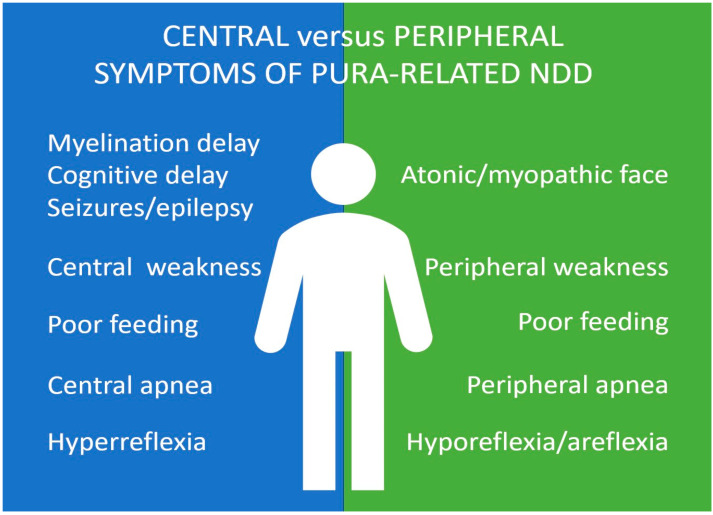
Graphical summary of central vs. peripheral symptoms in PURA-NDD.

**Table 1 ijms-24-02260-t001:** Summary of motor and neuromuscular features.

Yes	No	%
Hypotonia		
187	4	98
Respiratory		
124	20	86%
Myopathic face		
53	32	63%
Ambulation		
73	79	48%
Ptosis		
7	39	15%
DTR		
Abnormal	Normal	
17	3	
EMG/NCV		
Abnormal	Normal	
9	7	
Muscle biopsy		
Abnormal	Normal	
9	3	

DTR—deep tendon reflexes.

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
