# Peer review of "Neuromuscular and Neuromuscular Junction Manifestations of the PURA-NDD: A Systematic Review of the Reported Symptoms and Potential Treatment Options"

_ijms, 2023, doi:10.3390/ijms24032260_

Round 1

Reviewer 1 Report

This study reports the outcomes of a systematic review of the literature covering a rare condition: PURA-NDD, with a particular focus on potential involvement of neuromuscular transmission (NMT). The authors make the case that this is a neglected and potentially clinically very important site for generating clinical symptoms.

This is a careful, balanced and interesting review, making a compelling case that impaired NMT should be investigated further as a potential therapeutic target in these patients. The authors take care to highlight the limited and preliminary nature of much of the information available, and make recommendations with due caution. Thus, it is a considered and scientifically well-justified treatment of the subject that should be of great benefit to neurologists who might encounter such cases, and to the wider scientific community for raising awareness of this condition. 

I have only a few minor comments:

English and grammar comments:

There are relatively frequent minor errors of English and grammar, e.g.:

L242 - 'falled frequently' 

L243 - 'worsen in walking ability'

L306-307 - 'Even less children undergoing EMG studies and muscle biopsies.' This is neither a sentence, nor a meaning that is understandable in context.

Can text be revised by a fluent English speaker?

Scientific comments:

L71-74 - defects in hippocampus and cerebellum are reported. Were these selective (i.e. other regions showed no defects) or simply the areas studied, with other regions ignored?

L81 - the timespan of 5-15days postnally in rodents is when PURA involvement is intense. This is the period of synapse elimination at the NMJ. Can the authors add if this is relevant and worth investigating?

L135-7 - 'although electrophysiological studies suggest mixed pre- 136 and postsynaptic localization.' Please add reference(s) - either to other studies, or other sections of the review. This is an important point likely to be of interest to basic science and clinical colleagues.

L193 - 'regression of weakness'. Please clarify. It could imply weakness either progressively worsens, or improves.

L259-261 - can the fibre types affected be clarified? Is it systematic, or does fibre type disproportion differ in myosin isoform between cases?

L405-407 - should fluoxetine be added as a suggested line of treatment in the conclusion? L373-385 suggest it might be useful. 

Author Response

Dear Reviewer,

Thank you for your valuable consideration and for your time spent on our manuscript. Please find our detailed answers below:

Thank you very much for all your time and effort.

English and grammar comments:

L242 - 'falled frequently'

'had frequent falls'

L243 - 'worsen in walking ability'

'worsened walking ability'

L306-307 - 'Even less children undergoing EMG studies and muscle biopsies.' This is neither a sentence, nor a meaning that is understandable in context.

'In addition, in several reported cases, according to current, local standard of practices, invasive studies such as EMG and muscle biopsies have not been performed.'

Can text be revised by a fluent English speaker?

The manuscript has been reviewed by a native English speaker. The errors overlooked in the prior draft have been re-revised.

Scientific comments:

L71-74 - defects in hippocampus and cerebellum are reported. Were these selective (i.e. other regions showed no defects) or simply the areas studied, with other regions ignored?

From the referenced manuscript of Khalili et al. it seems that only these brain regions have been investigated. For the clarity we have updated this manuscript part as follows:

'Knockout Pura-/- mice were shown to have a significantly decreased neuron density in all three investigated regions of interest: cortex, cerebellum and hippocampus and decreased synaptic density in hippocampal neurons, suggesting a role in neuronal transport, synapse formation and memory formation [13]. There is no information regarding neuron and synapse density in other brain regions.'

and in L85-89:

'In the pathomorphological examination heterozygous Pur-alpha mice present with significantly reduced number of neurons and dendrites in the two out of four investigated brain regions: cerebellum and hippocampus. No significant differences have been reported for the amygdala and prefrontal cortex.'

L81 - the timespan of 5-15 days postnally in rodents is when PURA involvement is intense. This is the period of synapse elimination at the NMJ. Can the authors add if this is relevant and worth investigating?

We thank the Reviewer for this comment. We added the following information:

'Developmental NMJ synapse elimination in mice occurs during the first two weeks postnatally. In the sternomastoid muscle at birth 90% of cells were multiple innervated at birth, a little less than half of them already single innervated at P7 and all of them single innervated at [16].  Based on this information, the possibility of PURA’s involvement in pruning of NMJ synapses or synapse elimination as a functional role for PURA cannot be excluded, and in fact, is deemed more likely'.

L135-7 - 'although electrophysiological studies suggest mixed pre- 136 and postsynaptic localization.' Please add reference(s) - either to other studies, or other sections of the review. This is an important point likely to be of interest to basic science and clinical colleagues.

The reference to the study of Wyrebek et al. [34] has been added.

L193 - 'regression of weakness'. Please clarify. It could imply weakness either progressively worsens, or improves.

To provide clarity, we changed it to:  'During development, weakness associated with PURA syndrome usually mildly improves, especially after 1 year of life, however it may persist.'

L259-261 - can the fibre types affected be clarified? Is it systematic, or does fibre type disproportion differ in myosin isoform between cases?

Unfortunately myosin staining was reported only in two cases.

We clarified this as follows: 'Myosin staining was reported only in two cases: in one case, an increased neonatal myosin; and in the second case, mild atrophy of the fasts myosin-containing muscle fibers were observed'.

L405-407 - should fluoxetine be added as a suggested line of treatment in the conclusion? L373-385 suggest it might be useful.

We have added the following to provide clarity: 'Treatment with fluoxetine and other NMJ modulators may be considered in older children, as it is approved for children ages 8 and older'.

Reviewer 2 Report

Dear Authors,

This is a well-organized review with a crucial focus the presentation of Neuromuscular and neuromuscular junction manifestations of the PURA-NDD. The manuscript supports our knowledge about the motor symptoms of PURA-NDD, describe the neuromuscular phenotype and emphasize the role of potential treatment opportunities.

There are no major problems with this review.

Several things in the presentation need to be clarified and corrected.

- In itroduction section it is needed to add information about definition of Neurodevelopmental disorders (NDDs). There is also lack of PURA-NDD definition in the main text body.

- Table 1 should be reedited because it is not illegible.

- There are two Figures 1.

- Meaning of central and peripheral symptoms should be clearly discribe.

- Line 190: the phraze „No hypotonia was noted, only in four patients” is not understandable.

- Line 263 instead of myosin fibers should be muscle fibers.

This is an interesting and important manuscript. I recommend the manuscript for publication.

Author Response

Dear Reviewer,

Thank you for your valuable consideration and for your time spent on our manuscript. Please find our detailed answers below.

Thank you very much for all your time and effort.

- In itroduction section it is needed to add information about definition of Neurodevelopmental disorders (NDDs). There is also lack of PURA-NDD definition in the main text body.

The definition has been provided in the L45-47. It has also been added to the main body text, L194-196.

- Table 1 should be reedited because it is not illegible.

Table 1 has been edited.

- There are two Figures 1

Thank you. Figure numbering has been changed.

- Meaning of central and peripheral symptoms should be clearly discribe.

We thank the Reviewer for this remark. We clarify it as follows:

'Hypotonia is a general term used to describe a diminished tone and can result from the abnormality of the central nervous system (central hypotonia), peripheral neuromuscular system (peripheral hypotonia) or both (mixed hypotonia). Usually, it is thought that some symptoms can suggest the type of hypotonia. Impaired cognitive development, seizures and hyperreflexia feature central hypotonia, whereas decreased DTRs, lack of antigravity movement, muscle atrophy together with a normal cognitive function point toward peripheral hypotonia associated with neuromuscular weakness'.

- Line 190: the phraze „No hypotonia was noted, only in four patients” is not understandable.

This has been changed to: 'In four patients, no hypotonia was reported'.

- Line 263 instead of myosin fibers should be muscle fibers.

This has been changed to 'muscle fibers'